# SDAA: Secure Data Aggregation and Authentication Using Multiple Sinks in Cluster-Based Underwater Vehicular Wireless Sensor Network

**DOI:** 10.3390/s23115270

**Published:** 2023-06-01

**Authors:** Samuel Kofi Erskine, Hongmei Chi, Abdelrahman Elleithy

**Affiliations:** 1Department of Computer and Information Science, Florida Agricultural and Mechanical University, Tallahassee, FL 32307, USA; 2Department of Computer and Information Science, William Paterson University, Wayne, NJ 07470, USA; elleithya@wpunj.edu

**Keywords:** SDAA, MNA, UWSN, UVWSN, UV, MAC, CBND, USN, BS, trustworthiness/privacy, energy efficiency, GW

## Abstract

Security is one of the major concerns while designing robust protocols for underwater sensor networks (UWSNs). The underwater sensor node (USN) is an example of medium access control (MAC) that should control underwater UWSN, and underwater vehicles (UV) combined. Therefore, our proposed method, in this research, investigates UWSN combined with UV optimized as an underwater vehicular wireless network (UVWSN) that can completely detect malicious node attacks (MNA) from the network. Thus, MNA that engages the USN channel and launches MNA is resolved by our proposed protocol through SDAA (secure data aggregation and authentication) protocol deployed in UVWSN. SDAA protocol plays a significant role in secure data communication, as the cluster-based network design (CBND) network organization creates a concise, stable, and energy-efficient network. This paper introduces SDAA optimized network known as UVWSN. In this proposed SDAA protocol, the cluster head (CH) is authenticated through the gateway (GW) and the base station (BS) to guarantee that a legitimate USN oversees all clusters deployed in the UVWSN are securely established for providing trustworthiness/privacy. Furthermore, the communicated data in the UVWSN network guarantee that data transmission is secure due to the optimized SDAA models in the network. Thus, the USNs deployed in the UVWSN are securely confirmed to maintain secure data communication in CBND for energy efficiency. The proposed method is implemented and validated on the UVWSN for measuring reliability, delay, and energy efficiency in the network. The proposed method is utilized for monitoring scenarios for inspecting vehicles or ship structures in the ocean. Based on the testing results, the proposed SDAA protocol methods improve energy efficiency and reduce network delay compared to other standard secure MAC methods.

## 1. Introduction

Underwater wireless sensor networks (UWSN) and underwater vehicles (UV) can expand our scientific, commercial, and naval capabilities [1]. UWSN network exploration has received attention from numerous researchers [1]. However, security investigation for UWSN and UV, as one network entity, is unavailable based on state-of-the-art technology. Therefore, this research investigates combined UWSN and UV as optimized underwater vehicular wireless sensor networks (UVWSN), has proven useful and visually mapping benthic habitats and inspecting the structures of ships [2].

UVWSN network tends to automate underwater traffic, utilizing underwater vehicles. UVWSN can be utilized for ocean vehicle monitoring. The underwater acoustic network has increased nowadays. While much effort in the research of UVWSN has shifted, designing network security in UVWSN still needs to be thoughtfully complied with. An incredibly significant aspect is that UVWSNs are vulnerable to malicious node attacks (MNA). This is through the characteristics of underwater acoustic communication channels and the vehicles they deploy. Examples are the long propagation of signal delay, higher bit error rates, and a low communication bandwidth. All these network design challenges have a greater possibility of affecting UVWSN, which this research investigates.

Therefore, UVWSN must provide a higher speed data transmission rate with dependable, secure, and low latency metrics [3]. An essential feature of the UVWSN network is that it assists in efficiently monitoring and inspecting underwater applications, such as vehicles, including ship structures. However, the efficiency of the network may be possible after deploying the required security mechanism, including secure data aggregation and authentication (SDAA) optimized methods. SDAA-optimized methods include cluster-based network design (CBND) plans. In addition, SDAA methods provide the protection offered in the UVWSN through underwater sensor nodes (USN), which tends to isolate any MNA or vulnerable external nodes [4]. Another essential feature of the SDAA method is that the base station (BS) is employed to confirm the USN’s legitimacy after being deployed in the UVWSN. Therefore, BS awards trustworthiness/privacy protection through the authentication of each USN before they can be allowed to be part of the UVWSN.

Moreover, multiple mobile sinks (MMS) [5] have mobile nodes that utilize more energy efficiency mechanisms for convenient use in the network [5]. In addition, MMS controls the dynamic mobility of the USN [5]. This adjusts the MMS moving speed and trajectory while traveling in the underwater vehicle, reducing any MNA that may persist in the network. The detection of MNA in the SDAA methods is also due to its high energy-withholding capability.

In addition, the CBND method prolongs the lifetime of UVWSN, utilizing secure data aggregation and authentication methods that deploy MMS arrangements in the network. Moreover, CBND collects specific data from its members, including the cluster head (CH) and cluster node (CN) of USN in the UVWSN, which assist in organizing the underwater environment information and then forwards it to the MMS. The SDAA method compares design concerns with other secure medium access control (MAC) standards protocols that do not utilize CBND in their network deployment, as described below.

The authors in [6] proposed self-sustaining, efficient, and forward-secure cryptographic constructions for unattended wireless sensor networks (SEFSC). In addition, the authors in [7] proposed the synopsis diffusion approach (SDA), and the authors in [8] proposed the energy-efficient and secure transmission scheme based on chaotic compressive sensing in underwater wireless sensor networks (EEST). The authors in [6,7,8], proposed protocols that were investigated based on secure MAC UWSN, but they did not utilize CBND. Moreover, the authors in [9] proposed authentication methods based on secure MAC UWSN. However, all these proposed secure MAC protocols are independent of CBND deployment, which did not include vehicle deployment scenarios in the UWSN.

Since USN (underwater sensor node) depicts a medium access control (MAC), their credentials in UVWSN, which utilized our proposed SDAA method, should be stored in nearby BS in the CBND method.

In addition, deploying only underwater wireless sensor networks (UWSN) in secure MAC protocols such as SEFSC, SDA, and EEST, which do not utilize multiple mobile sinks (MMS), is observed as a design challenge. One crucial design advantage of our proposed UVWSN is that MMS [5] deployment in the network assists in making better decisions for appropriately planning the moving path of vehicles, including ships, in the ocean. This helps to avoid any overhead information [5] exchange with each USN deployed in the network. Therefore, the entire network architectures proposed in secure MAC protocols, including [6,7,8], were vulnerable and exposed to malicious node attackers (MNA).

Consequently, this leads to higher MNA encounters in the network, which also leads to stealing or altering information about their networks’ stored data credentials. In addition, this compromises the trustworthiness/privacy and integrity of the data concerns in the UWSN used in [6,7,8].

However, underwater sensor nodes (USN) deployment in the UWSN network requires sensitive credentials, such as locations and identities. Otherwise, this enables the encounter of a malicious node attacker (MNA) to reveal the location of the physical access of the USN, and this situation could expose the entire network to be compromised. Moreover, it is noteworthy that the identifications (ID) of the sensor nodes (SN) deployed in [6,7,8] used different encryption and decryption methods. Based upon this, the SN credentials must require secure storage mechanisms to preserve the data’s trustworthiness/privacy and integrity. However, the limitations of trustworthiness/confidentiality and data integrity could destroy the deployment of the SN since MNA could steal the location and make it impossible for the UWSN to access data securely in SN.

To detect and remove malicious nodes from the UWSN deployed in secure MAC protocols such as SEFSC [6], SDA [7], and EEST [8] network and to preserve trustworthiness/privacy, many studies have investigated trustworthiness/privacy provisions for UWSN. These studies depend on the authentication of underwater vehicles only in [9,10,11,12]. However, the studies that rely on detecting malicious nodes can be found only in [13,14], which depend on the trustworthiness/privacy provision in UWSN.

Therefore, the need to propose a new underwater sensor network, such as the proposed SDAA protocol, to investigate all the design challenges in secure MAC protocols, including SEFSC [6], SDA [7], and EEST [8], which should utilize CBND, and underwater vehicle system, should be a priority. Therefore, our proposed SDAA method achieves that design requirements and detects and prevents malicious node attackers (MNA) using UVWSN with a sense of urgency [15].

In addition, our proposed SDAA method introduces secure medium access control (MAC) through USN, which utilizes a protocol design that provides trustworthiness/privacy in the UVWSN (underwater vehicular wireless sensor network).

Therefore, in this paper, we propose a secure MAC protocol, known as the secure data aggregation and authentication (SDAA) optimized methods, that employ MMS (multiple mobile sinks) in cluster-based network design (CBND) that ensures trustworthiness/privacy provision in UVWSN. The application deployment of our proposed UVWSN is for monitoring and inspecting vehicles and ship structures for navigational systems in the ocean. Based upon this, we can reevaluate new performance metrics, such as reduced delay, reliability, packet drop, and energy efficiency in the proposed new UVWSN.

The main contributions of this paper are as follows:We propose a new secure SDAA method that employs MMS (multiple mobile sinks) in the cluster-based network design (CBND). MMS, including ships, is used for appropriate vehicle path planning and avoids communication overhead. Comparatively, secure MAC protocols, such as SEFSC, SDA, and EEST, do not utilize MMS and CBND in their networks; therefore, the energy efficiency for the network is an issue.The proposed SDAA protocol method minimizes latency/delay, providing reliability and ensuring energy efficiency provision in USNs (underwater sensor nodes) that improves packet delivery ratio, and reduces packet drop in the network, as compared to secure MAC protocols SEFSC, SDA, and EEST which only detects packet drops and delay, without accounting for any energy efficiency in the network.We propose a new underwater vehicular sensor network (UVWSN) involving underwater vehicle (UV) deployment in the network that can detect and prevent all forms of malicious node attacks (MNA), for practical wireless sensor monitoring applications for the underwater vehicle (UV), as compared with secure MAC protocols, including SEFSC, SDA, and EEST, in which only UWSN was used to assess malicious attacks, without accounting for any UV.Therefore, the proposed UVWSN ensures complete trustworthiness/privacy and integrity provision in the network, using trusted encryption–decryption schemes that improve security, compared to other secure MAC protocols, including SEFSC, SDA, and EEST, which only provide privacy without any data integrity provision in their network.The proposed SDAA protocol includes a cluster-based network design (CBND) that improves energy efficiency and prolongs the network lifetime of the USN for more extended application deployment. Secure MAC protocols, such as SEFSC, SDA, and EEST, use only authentication. However, it is not a cluster-based network (CBND); therefore, it is difficult to account for actual energy efficiency deployment in the UWSN.

The remainder of the paper is as follows: Section 2 presents the related work of various secure MAC protocol classes; Section 3 presents design challenges of secure UWSN MAC protocols and models of secure data aggregation and authentication using UVWSN; Section 4 shows simulation setup and experimental results including discussion. Section 5 concludes the entire paper as follows.

## 2. Related Work

MAC (medium access control) protocol classes can complete a common task in WSN data transmissions but do not provide a secure channel in data transmissions. This section details MAC protocol-related works, subdivided into two sections, including Section 2.1: Contention-based MAC and Section 2.2: Contention-free-based MAC, as follows.

### 2.1. Contention-Based MAC Protocols

The authors in [15] proposed a contention-based MAC protocol that enhances the system performance of an underwater acoustic sensor network. The proposed MAC protocol employs a widely popular machine-learning technique using only Q-learning. The proposed protocol allows a sensor node that intelligently selects back-off slots and schedules data transmission in such a way that it minimizes collision without taking care of the vulnerabilities of the MAC. However, the authors in [15] admitted that the sensor nodes are not required to exchange scheduling information. This presupposes that the proposed protocol may include low complexity and high overload.

In another development, the authors in [16] estimated that contention-based underwater MAC protocols could utilize total underwater channel bandwidth. However, the protocols do not provide any means of secure data channels in communication. In addition, two classifications of contention-based MAC protocols are handshaking and random-access MAC. Handshaking-based MAC protocols utilize sender and receiver nodes, and they can exchange control packets, before enabling data transmission, for channel reservation and collision avoidance. Thus, in [16], there is no secure processing of data transmission used in the network, which may lead to trustworthiness concerns.

Random access-based MAC protocols, however, utilize probabilistic approximation in collision avoidance. The reason is that there is no prior coordination in the sender data transmission. This presupposes that trustworthiness provision is a design concern in the network.

Therefore, handshaking-based MAC protocols are dominant over random access-based protocols. This assumption can be possible since the protocols incur collision avoidance and insecure channel data processing, which leads to energy inefficiency, and introduces latency/delay in their network.

The authors in [17] proposed a handshake-based MAC protocol design analysis with delay variation in underwater acoustic networks. The proposed MAC protocol aims to identify an essential characteristic of two-scale delay variation based on field test results. Based on this, we performed a simulation to study the impact of delay variations in the MAC of the network. The results suggested a slot length adaptation scheme for handshake and slotting-based MAC protocols. This protocol, also known as slotted-FAMA, has an adaptive slot length estimated to achieve more throughput.

The authors in [18] proposed a protocol based on slotted FAMA. This protocol used only adaptive slot duration to avoid packet collision, using a handshaking technique, without providing any secure method.

The authors in [19] proposed a protocol based on cooperative authentication in underwater acoustic sensor network MAC protocol. This proposed protocol includes a new algorithm for message authentication in UWSN. However, the protocol behavior mimicked the channel without associated legitimate data transmission.

### 2.2. Contention-Free-Based MAC Protocol

Contention-free MAC (CFM) protocol brings the network from an arbitrary state to a collision-free stable condition. A significant concern worth investigating is the distribution process of the contention-free MAC protocol. The steady state of contention-free MAC protocol transmits messages without collision. The metrics used in a stable condition include the efficiency of the node’s throughput and the inverse time interval between when messages reach the network.

In [20], the authors proposed an energy-conserving collision-free MAC protocol for underwater sensor networks. The proposed contention-free MAC protocol indicates reliability in the network.

The authors in [21] proposed a collision-free coloring MAC protocol for underwater sensor networks (UWSN). The proposed protocol aims to achieve high throughput performance by avoiding collision at the MAC layer. The protocol is to improve energy efficiency and packet delivery fairness.

These network design limitations concerning secure MAC protocol authorize cluster-based network design (CBND) of the individual sensor nodes. Furthermore, the network requires underwater vehicle (UV) investigation in the underwater wireless sensor network (UWSN) deployment.

Therefore, our proposed SDAA protocol fulfills all these design limitations requirements and utilizes vehicle deployment in the UWSN. Our proposed SDAA protocol optimizes the network to be UVWSN, which uses secure data aggregation and authentication in CBND methods. Based upon this, we can reevaluate the reliability, energy efficiency, and packet drop in the network and compare it to other secure standard MAC protocols, such as SEFSC [6], SDA [7], and EEST [8].

## 3. Design Challenges of SEFSC, SDA, and EEST Protocols and Models of Secure Data Aggregation and Authentication Using UVWSN

In this section, we highlight the design challenges of UWSN secure standard MAC protocols, including SEFSC [6], SDA [7], and EEST [8]. We also compare it with our proposed SDAA secure MAC protocol methods.

SEFSC is a secure MAC underwater wireless sensor network (UWSN) MAC protocol, which has operated in hostile underwater conditions since it is deployed in that condition. Based on this, the protocol experienced more significant security challenges in the network. Therefore, the protocol network performance should depend on its collected mobile data. However, due to the channel unreliability nature of UWSN operation in hostile conditions, the protocol needed to achieve reliability of mobile data collection in CBND. In addition, the UWSN, which they investigated, was also vulnerable. Therefore, we anticipate malicious node attacker detection for mitigating channel unreliability challenges in the network. These channel unreliability challenges and hostile node encounters lead to communication overheads in the UWSN. However, SEFSC employed mobile data transmission to investigate UWSN channel operation conditions. Notably, multiple mobile sinks (MMS), which can appropriately plan mobile data routes, are needed to overcome communication overhead in the network. Our proposed SDAA protocol deployed in UVWSN employs MMS and overcomes these challenges.

SDA is a synopsis diffusion approach that is a secure MAC protocol. This protocol solves the challenge of data integrity, which means the protocol needs accountability for the trustworthiness of data or reliability in the UWSN. However, the SDA protocol could not detect malicious node attackers (MNA) in the UWSN. Even though the SDA protocol network architecture deployment utilized a base station (BS), the BS was the only authentication method in the network. Consequently, the SDA protocol used the BS yet could not efficiently detect any MNA. Another design challenge with the SDA protocol was that the protocol only noticed a few MNA. This presupposes that the protocol deployment in the UWSN was on a small scale. Therefore, scalability is a concern, leading to more channel insecurity in the UWSN deployment.

In all these network design challenges, SDA protocol leads to the unreliability nature of the USN (underwater sensor node) or the channels. Based on this, the USN channel’s packet drop and energy efficiency become more challenging. However, our proposed SDAA protocol depends on a clustered-based network design (CBND) that can reorganize the USN channel very well in a more energy-efficient manner. In addition, our proposed SDAA method utilizes multiple mobile sinks (MMS) for planning the appropriate path of underwater vehicle movement. Our proposed SDAA protocol reduces any communication overhead encountered in UVWSN. In addition, the challenge of the packet drop can be resolved through our proposed SDAA protocol using optimized secure data aggregation and authentication method deployed in CBND, which reorganizes the USN channels for greater energy efficiency and reduces packet drop in the network.

EEST secure MAC protocol is an energy-efficient and secure transmission scheme based on chaotic compressive sensing. The development and implementation of EEST aimed to guarantee data security for the USN, or channels, to prolong the UWSN lifetime. EEST used compressive sensing and exploited sensor data spareness using the time domain. Therefore, the protocol used a long-time transmission period. Thus, delay in the network was a pronounced effect, leading to energy efficiency limitations. EEST protocol reduced the number of transmissions of the USN or channel data during data sampling in each frame period, eventually transmitting the data to reduce the energy consumption of each USN in the UWSN. EEST chaotic compression encryption is used only to encrypt data at the end of specific periods.

This presupposes that delay and subsequent packet drop will be an issue. Another limitation is that the EEST protocol does not utilize MMS, and the UWSN architecture deployment was not in CBND.

To solve these challenges in EEST, our proposed SDAA protocol employs authentication and secure data aggregation optimized methods. It deploys the CBND network which utilizes MMS to ensure there is no communication overhead in the USN or channel and that makes the reliability of data transmission in UWSN is possible.

To solve all these design challenges encountered in the secure MAC protocols, such as SEFSC, SDA, and EEST, our proposed SDAA protocol develops a new secure system model and probability analytical models to reevaluate new performance metrics for the reliability, average packet drops, and energy efficiency of the UVWSN.

### 3.1. Secure Data Aggregation and Authentication Modeling

Secure data aggregation and authentication ensure network reliability and energy efficiency are maintained in the UVWSN in CBND. Therefore, due to the unreliable nature of the USN channel and packet drop deployment found in secure MAC protocols, including SEFSC, SDA, and EEST, our proposed SDAA protocols employ secure data aggregation and authentication-optimized methods based on cluster-based network design (CBND). Based upon this, we modeled a cluster network in [22] for our proposed SDAA protocol to evaluate data reliability and compare the performance of the network deployment with the other secure MAC protocols, EEST, SDA, and SEFSC.

Therefore, our proposed SDAA protocol employs optimized secure data aggregation and authentication methods to use trustworthiness or data integrity to provide data reliability deployment in CBND for UVWSN Consequently, we can deploy UVWSN to monitor and inspect vehicles, including ship structures [23]. For the proper authentication of individual USNs (underwater sensor nodes) in the network, the SDAA protocol utilizes the base station (BS). It deploys cluster heads (CH) and cluster nodes (CN) in the UVWSN.

Figure 1 depicts the secure data aggregation and authentication optimized arrangement deployment in CBND that utilizes two steps: The first involves labels 1–4. These steps involve the aggregation process. In step 1, CH sends a registration request to the gateway. In step 2, the request is decrypted and retrieved, and in step 3, the GW generates the hash value of the data. In step 4, the decision of the hash value is determined by CH, if it is yes or no. Secure authentication of cluster heads (CHs) follows. The authentication process also utilizes protected data aggregation (found in the next section).

In the second step, which involves labels 5–9, gateways (GWs) authenticate CHs. The authentication process is required to confirm that the operating CH in each cluster is a valid USN. Based upon these steps, the CH node must also not be compromised. This ensures that the entire cluster is in safe operation. Each USN is used to protect the data transmissions during the authentication process. It uses symmetric encryption (shared key encryption process) and sends the data to the CH. The data are then aggregated securely and transmitted to the base station (BS). Timestamp values are processed whenever compromised data or MNA (malicious node attacks) are detected, confirming the secure network operation.

### 3.2. System Model

The system model of the proposed SDAA protocol in the UVWSN aims to solve design challenges, including reliability, packet drop, and minimizing delay in UVWSN data transmission, based upon the CBND process, which we introduced by modeling [23]. Therefore, the system model employs secure data aggregation and an optimized authentication method deployed in similar conditions to the other secure MAC protocols, including SEFSC, SDA, and EEST, without cluster-based network design (CBND) deployment. Based upon this, the SDAA protocol deploys multiple mobile sinks (MMS) (which will be explained in algorithm 4 in the next section) in CBDN. Our proposed SDAA protocol also deploys secure data aggregation and authentication-optimized methods to evaluate the network design challenges, based on the communication overhead, due to malicious node attacks (MNA) in the UVWSN. Therefore, our proposed SDAA protocol utilized the USNs (underwater sensor nodes) or channels deployed in the UVWSN. However, UVWSN has peculiar features of underwater acoustic channels that use low signal propagation speed (approximately 1.5 km/s).

Therefore, the system model utilizes USN, CH (cluster head), GW (gateway), and base station (BS). Consequently, the system model uses UVWSN transmission monitoring techniques. Figure 2 depicts the proposed system model. Each USN involves at least one cluster managed by CH, and BS associates with CH to the GW through acoustic links. Acoustic links apply to the UVWSN monitoring application. Thus, GW has unlimited energy resources because of optimized secure data aggregation and authentication methods, which use perfect time synchronization of the USN information delivery in the network. The complete energy efficiency capability of the system model is also due to the multiple mobile sinks (MMS) and CBND method deployed in the proposed SDAA protocol. This can reduce overhead information in the network, thereby increasing energy efficiency. During the monitoring operation of the UVWSN application, two or more GWs can link together. This reduces any delays in the network. GWs can communicate with each other through radio frequency (RF) links.

### 3.3. Secure Authentication of Cluster Head

Secure authentication of cluster head utilizes authentication and secure data aggregation optimized methods to authenticate each USN or cluster node (CN) in the UVWSN. Therefore, cluster heads (CHs) [23] are a gateway between UNS and BS. CH deployment is vital in securing data aggregation and authentication in the UVWSN.

In addition, through the secure authentication of CH, route establishment among clusters and all the CHs are authenticated to a GW. Here, every CH in the network initially generates a secret key and sends a registration request with a GW. CH further creates a hash value and signs it. Thus, CH uses a secret key and sends a request to another GW. The request is decrypted at GW using the public key of CH. CH also retrieves the time stamp. GW generates registration confirmation using the hash value received, including the time stamp. Both hash values are compared. Whenever any variation is observed, it is ignored, and this further authenticates CH. GW sends a registration response to CH after signing the hash value. Thus, CH uses the secret key of GW. Subsequently, CH determines that it has been authenticated. It does that by decrypting the registration response received using the public key of a GW. The process of CH authentication is also described in Algorithm 1.
**Algorithm 1:** Cluster head authentication process1. Every *CH* initially generates *SK_CH_*.*2. CH* creates *CH_RR_* which consists of [*CH_id_*, *GW_id_*, *T*_*s*1_]*3. CH* signs *T*_*s*1_ with *SK_CH_* to create *SK_CH_* (*T*_*s*1_)*4. CH* generates a hash value of *CH_RR_* as *H*_1_ = *H* (*CH_id_*, *GW_id_*, *T*_*s*1_)*5. CH* generates *M_RREQ_* by signing *H*_1_ using *SK_CH_* as *M_RREQ_* = *SK_CH_* [*H* (*CH_id_*, *GW_id_*, *T*_*s*1_)]  6. *CH* sends *SK_CH_*(*T*_*s*1_) and *M_RREQ_* to *GW*  7. *GW* decrypts *SK_CH_*(*T*_*s*1_) using *PK_CH_* and retrieves *T*_*s*1_  8. *GW* decrypts *M_RREQ_* using *PK_CH_* and retrieves *CH_RR_*  9. *GW* creates hash value of *CH_RR_* as *H*_2_ = *H* (*CH_id_*, *GW_id_*, *T*_*s*1_)  10. *GW* compares *H*_1_ and *H*_2_    If *H*_1_ does not match with *H*_2_,       Then, *GW* discards *M_RREQ_*      Else *GW* confirms that *C**H* is a valid node.      *GW* creates as *CHR_C_* [*CH_id_*, *GW_id_*, *T*_*s*2_]      *GW* creates hash value of *CHR_C_* as *H*_3_ = [*H*(*CHR_C_*)]      *GW* signs [*H* (*CHR_C_*, *T*_*s*2_)] using *CK_GW_* to create *M_RRES_* as           *M_RRES_*= *SK_CH_* [*H*(*CHR_C_*)]      *M_RRES_* is sent to *C**H* by *GW*
    End If  11. *C**H* decrypts it using *PK_GW_* and retrieves *H* (*CHR_C_*, *T*_*s*2_)  12. *C**H* determines that *GW* has authenticated it

Following, a GW individually and securely authenticates every CH. This ensures that any vulnerable or malicious node attacker (MNA) encountered in the UVWSN does not control the cluster operation, protecting the cluster from being compromised. Table 1 shows the used notation of the proposed secure authentication algorithm.

### 3.4. Protected Data Aggregation

Protected data aggregation [24,25] occurs after ensuring that the selected CH is well-authenticated as a feasible solution for the UVWSN.

This ensures that each USN can transfer data and securely aggregate it to its CH operation. Each USN in the UVWSN deployment is controlled by protected data aggregation. Thus, each USN protects the data transmission using symmetric encryption before sending it to the CH. The data transmission at CH is aggregated securely and further transmitted to base station (BS), where compromised MNAs (malicious node attackers) can be detected, removed, and managed accordingly. This process is described in Algorithm 2.

Algorithm 2 ensures that secure data transmission is well performed in the network. The secure methods used in the algorithm include secure authentication and secure data aggregation by each CH in the UVWSN. Therefore, BS checks the aggregated data for authenticity, and BS checks its time stamp as well. Any detected compromised USN data are discarded, and this ensures the safety of the remaining aggregated data. The detected compromised/malicious node is isolated from the cluster. This maintains network security in the network.

A process flow diagram of the proposed SDAA protocol is used and displayed in Figure 3 to show the overall processing of information. In Figure 3, process flows 1 and 2 indicate the CH authentication with its GW. Process flows 3 indicates the symmetric data encryption by the USN, and process flow 4 demonstrates the decryption of received data by the BS. Used notations are shown in Table 2.
sensors-23-05270-t002_Table 2Table 2Used notations and description for Algorithm 2.NotationsMeaningSiSensorsSidID of SiTs3*Timestamp at which the data are sensed at S**K_m_**Master key**K_i_**Encryption key of S_i_**K_d_**Decryption key of BS**E_ncK_**Encryption using* *K_i_**D**Sensed data**D_Enc(i)_**Encrypted data at* *S_i_**D_Enc(col)_**Collection of encrypted data from all* *S_i_* *at CH**D**_Enc_*(*_CH_*)*CH’s encrypted data**D**_Enc_*(*_agg_*)*Aggregation of encrypted data*
**Algorithm 2**: Secure data aggregation process1. GW initially generates *K_m_* and then builds *K_i_* for each Si in the cluster using H as           *K_i_*= H (*K_m_* || Si)  2. While transmitting D, Si builds a hash value HMAC as         HMAC = MAC (D || Ts3)  3. *S_i_* encrypts *HMAC* along with *D* and sends it to respective *C**H*.  4. *D_Enc(i)_* = *E_ncK_* [*D* || *HMAC* || *S_i_d*]5. *C**H* collects *D_Enc(i)_* from all *S_i_* and creates *D**D_Enc(col)_* as     *D**_(col)_* = *D**_Enc_* (1) + *D**_Enc_* (2) + … + *D**_Enc_*(*n*) because *i* = 1, 2…, *n*6. *C**H* aggregates *D**D_Enc(col)_* with its own encrypted data *D**_Enc_*(*C**H*) and send it to *BS*  7. *D**_Enc_*(*_agg_*) = *D**D_Enc(col)_* + *D**_Enc_*(*_CH_*) 8. *BS* decrypts *D**_Enc_*(*_agg_*) using *K_d_* and retrieves *D* and *T_s_* sent by each *S_i_*.9. *BS* compare *T_s_* associated with each *S_i_*  10. If any *T_s_* is found older than others,  11. Then, the associated *S_i_* is confirmed as malicious, and *D* is discarded.

### 3.5. Probability Analytical Model for Malicious Node Attacker(MNA) Detection

This research utilizes a probability analytical model for detecting malicious nodes attackers (MNAs) in the underwater vehicular wireless sensor network (UVWSN), utilizing USN and other network devices. Therefore, using the probability analysis of the proposed SDAA protocol, we determine possible MNAs in UVWSN, deployed in a comparable situation in the other secure MAC protocols, including SEFSC [6], SDA [7], and the EEST [8]. Therefore, we model packet drop in [6], using the probability analysis for detecting MNA to correct any signature mismatch in the network. Subsequently, we determine the network reliability and packet drop.

The probability of false detection and isolation of MNA, in our proposed SDAA protocol, becomes necessary and is modeled as follows:
In the probability analytical model for MNA detection, the following parameters are used:$P_{r}$ = Probability of detecting malicious USN due to mobility and other external factors$P_{D}$ = Probability of packet drops in the networkR = Packet generation rate of the USN deployment in UVWSNt = Time interval during data transmission and secure data aggregationSMi = Number of signature mismatches of node $N_{i}$ among its neighbors

Then, the probability of SM exceeding the maximum threshold ${SM}_{th}$ is given by Equation (1).
(1)$$P_{r}\left(SM>{SM}_{th}\right)=1-\sum_{i=1}^{{SM}_{th}}\left(\begin{array}{c} Rt\\ i \end{array}\right)P_{D}^{i}{\left(1-P_{D}\right)}^{Rt-i}$$
where,
*W* = number of warnings received by node *N_i_* when any of its neighbors turns to be malicious or vulnerable nodes.The probability of *W* exceeding the maximum threshold $W_{th}$ is given by Equation (2).
(2)$$parent\;P_{r}(W\left(N_{j}\right)>W_{th}=1-\sum_{i=1}^{W_{th}}\left(\begin{array}{c} NH\\ i \end{array}\right)P_{FD}{\left(N_{j}\right)}^{i}{\left(1-P_{FD}{(N}_{j})\right)}^{NH-i}$$
where,
*N**H* is the number of neighbors of node $N_{i}$$P_{FD}$ ($N_{j}$) is the probability of false detection of node $N_{j}$, which is given by Equation (3).
(3)$$P_{FD}\left(N_{j}\right)=1-exp\left[-2.\frac{{\left(RtP_{D}-{SM}_{th}\right)}^{2}}{Rt}\right]$$

Then, the probability of false isolation $P_{FI}$ is given by Equation (4).
(4)$$P_{FI}\left(N_{j}\right)=1-exp\left[-2.\frac{{\left(NHP_{FD}\left(N_{j}\right)-W_{th}\right)}^{2}}{NH}\right]$$

### 3.6. Secure Aggregation of Cluster Head Algorithm and Secure Sink Authentication

#### 3.6.1. Secure Aggregation of Cluster Head Algorithm

Algorithm 3, seen below, is implemented in our proposed SDAA protocol. This algorithm was deployed for secure aggregation by the cluster head (CH) and BS in similar conditions with the other secure MAC protocols, including SEFSC, SDA, and EEST, which were deployed without cluster-based network design (CBND). Therefore, the SDAA protocol utilizes optimized secure authentication and securely aggregates a substantial number of underwater sensor nodes (USN), ensuring scalability, utilizing single-hop secure data aggregation in the CBDN method. Based upon this, we can reevaluate the reliability, delay, and energy efficiency based on Algorithm 3.
**Algorithm 3**: Process of Secure Aggregation and Cluster Head (CH) and BS Single-hop Communication1. **If** CH sends a single-hop broadcast message to all cluster members and authenticates the sensor nodes, **then**2.  the cluster members use single-hop to receive the query (Q) message, and each sensor node receives information from the CH after authentication.3.  for CHSecureM=MSecureQ, then4.  CHSecureM sends and stores secure information to the BS after authentication, **then**5.   secure data aggregation nodes compute MAC single-hop, **then**6. secure aggregated nodes authenticate the secure aggregated data before sending it to BS nodes, **and**7. secure aggregated nodes send authenticated data to BS8. **End if**

#### 3.6.2. Secure Sink Authentication

Algorithm 4 is implemented in our proposed SDAA protocol that helps in secure multiple sink deployment, with authentication, and secure data aggregation optimized method, for verifying the legitimacy of new nodes that are deployed, based upon multiple mobile sinks (MMS) nodes in the network. The process is performed with the base station (BS) and the CH in clustered-based network design network (CBDN) methods. Our proposed SDAA protocol is deployed in a similar condition in which other secure MAC protocols, such as SESFC, SDA, and EEST were deployed; however, without CBND and MMS methods in the network. Therefore, in our proposed SDAA protocol, we utilized MMS and optimized protocol [26,27] and deployed it in the system model, including secure data aggregation and authentication in CBND. Based upon this, we reevaluate performance metrics for reliability and energy efficiency of the UVWSN network based on Algorithm 4 as below.
**Algorithm 4**: Secure Multiple Sink Authentication      BS: base station; ${NM}_{s}$: number of Mobile sinks; CH: cluster      head nodes; Cert: certificate of the Mobile sink; ${MM}_{i}$: mobile information message.      ${MS}_{q}$: query messages for CH, $B_{s}$, and ${MS}_{n}$; $A_{p}$:      Approval; ${MS}_{n}$: Mobile sink in the network; ${SD}_{p}$: secure data packets      $D_{L}$: distance between mobile sink; T: Broadcast time of Mobile nodes;      ${VM}_{s}$: valid mobile sink; ${LM}_{s}$: legitimate mobile sink;      $M_{si}$: illegitimate mobile sensor node; $EM_{n}$: entry into the network       1. Input ${(MM}_{i},\;{NM}_{id},\;{NM}_{s},\;T,\;D_{L})$
      2. Output $\left(A_{p},\;Cert\right)$      3. **for** each member of the mobile sink in             the SDAA models **do**      4. CH transmits a query message to       all its cluster member mobile sink node $\left({MS}_{q}\right)$
      5. after receiving $\left({MS}_{q}\right)$, the mobile sink node computes           ${MS}_{q}({CH}_{id}∥{MS}_{n}∥{MM}_{i}∥T)$
          6.       $CH\to {MS}_{q}$
          7. **If** any new mobile sensor node or mobile sink joins the network          8.  ${NM}_{s}\to {EM}_{n}\in {MS}_{n}$ **then**          9.    CH update $B_{s}$          10.      $B_{s}$ recall ${NM}_{s}$          11. $B_{s}$ investigate $Cert({NM}_{id},{SD}_{p},T,D_{L})$ for the ${VM}_{s}$          12.  $set\;A_{p}\;for\;{NM}_{s};\;A_{p}=(B_{s},\;{MM}_{i},\;{SD}_{p},\;T,\;D_{L},\;{NM}_{id})$
          13.     $B_{s}$ check broadcast $A_{p}({MS}_{q})$ with CH          14.      **If**
${NM}_{s}=={VM}_{s}={LM}_{s}({NM}_{s})$ **and**          15.        ${{NM}_{s}==MS}_{q}$ **then**          16.   $B_{s}$ approval ${NM}_{s}=={VM}_{s}{\in MS}_{n}$
          17.  **Else**          18.    ${NM}_{s}\neq V_{ms}\;and\;{NM}_{s}\neq {MS}_{q},\;{MS}_{i}={VM}_{s}$      19.   $B_{s}\neq {NM}_{s}$ and not approval of ${NM}_{s}$
   20.  **End if**   21. **End if**   22. **End for**

### 3.7. Modeling for Secure Underwater Vehicle (UV)

One of the main objectives of the proposed SDAA protocol is to use the protocol to investigate secure underwater wireless sensors (UWSN) utilizing underwater vehicles (UV) [27]. Since the deployment of UWSN combined with UV results in an underwater vehicular wireless sensor network (UVWSN), the primary responsibility of UVWSN is to deploy the network for monitoring and inspecting vehicles in the ocean, including ship structures, and for navigational purposes.

Therefore, our proposed SDAA protocol was deployed for evaluating the underwater vehicular condition of the UVWSN in a similar condition with other secure MAC protocols, including SEFSC, SDA, and EEST, which were deployed only utilizing UWSN and did not utilize any vehicle (UV). Therefore, our proposed SDAA protocol, which utilizes UVWSN involving the deployment of UV, can detect malicious node attacks (MNA) under similar conditions to the other secure MAC protocols, including SEFSC, SDA, and EEST. Consequently, we assessed new performance based on the reliability of the UVWSN for detecting MNA using our proposed SDAA protocol and compared it with SEFSC, SDA, and EEST under the same monitoring conditions and deployment for the inspection of ship structures.

Therefore, in our proposed SDAA protocol, we deployed USNs in the UVWSN by optimizing [8] and utilized this in the system model, such that UNS is placed in the rightful position for obtaining the USNs’ data reading in the network. Below, our novelty proposed mathematical model for UVWSN, for evaluating MNA data using the USN, without relying on navigational GPS location, is defined in Equation (5) as below:(5)$$\epsilon_{i}=\sum_{i=1}^{N}T_{in}+\frac{\Delta_{in}}{N}+N_{oin}$$
$T_{in}$ denotes the different output positions of different N mobile and static USN.$\frac{\Delta_{}}{N}$ denotes varying times unfairness on different periods.$N_{oin}$ shows the probability of MNA vulnerabilities present in the system measured.

Equation (5) is used for determining MNA in USN data that could be present during various average times of the UVWSN monitoring operation. This improves performance metrics in measuring the reliability of the USN data in the UVWSN. Moreover, reliability in the network leads to assessing the overall energy efficiency of the proposed SDAA protocol, as explained below.

### 3.8. Secure Data Energy Evaluation Modeling Method

Secure data energy efficiency of the network can be evaluated by our proposed SDAA protocol and is used to determine energy efficiency in a comparable way in which the energy efficiency of secure MAC protocols, including SCFSC, SDA, and EEST, was determined. This is possible to assess the overall performance metrics of the network, based on the reliability and reduced delay of the entire network, for the measurement of the energy efficiency of the network.

To comply with the performance metrics measurement in the entire network, our proposed SDAA protocol, therefore, optimizes the energy efficiency method used in [28] to evaluate the performance metrics of the network.

Based upon this, the energy consumption of the proposed SDAA protocol is divided into two parts: the first is secure data energy consumption $E_{SecureCollect}$, and the second is secure data transmission energy consumption $E_{SecureSend}$. Consequently, we define the secure total energy consumption as:(6)$${E_{TotalSecure}=E}_{SecureCollect}\;E_{SecureSend}$$
Here, $E_{SecureCollect}=xP_{SecureCollect}\cdot T_{SecureCollect}$ is the energy of secure energy consumption of authenticated and aggregated data collected in the CBND, $P_{SecureCollect}$ is the secure energy consumption of authenticated and aggregated data collected, and $T_{SecureCollect}$ denotes the secure time collection of x data bits of the authenticated and aggregated data in the network.

In addition, the energy consumption of secure data transmission is influenced by bandwidth, transmission loss, and transmission delay. Secure data transmission energy consumption comprises of two parts: secure transmission energy consumption due to vulnerability or malicious node attacker (MNA) detection. The energy consumption of the secure data transmission is defined based on [4,23,28]:(7)$$E_{SecureSend}=P_{SecureSend}.T_{SecureSend}.L(d)$$
Here, $P_{SecureSend}$ is the energy of the secure data transmission by the legitimate USN and vulnerable nodes in the UVWSN, $T_{SecureSend}$ is the time of secure data transmission, L(d) is the distance between CH and legitimate USN and the vulnerable sender and receiver underwater sensor nodes (USNs) energy loss, and d is the distance between the sender and receiver of the USN utilized in UVWSN.

In this research, the UVWSN network comprises the USN of the target legitimate USN and vulnerable nodes that securely collect the data through secure data aggregation and authentication methods deployed in the proposed SDAA protocol. This securely transmits data in the network using multiple mobile sinks (MMS) nodes, utilizing radio links in the CBND.

Since the speed is very rapid in underwater monitoring conditions, the secure data transmission time that utilizes the UVWSN monitoring application, which deploys MMS, CH, and BS, should be given priority. This will assist in measuring accurate energy efficiency and reduce delays in the network while utilizing USN links. This process is securely protected through the authentication of the secure aggregation methods in the network.

Consequently, different delays experienced in the network lead to obtaining accurate secure data collection delays, which include secure a computation delay $T_{SecureCompute}$ and time $T_{SecureUVS}$ used in the UVWSN network that involves target legitimate USN and vulnerable sensor nodes during secure data collection and waiting time $T_{w}$, determined as:(8)$$T_{SecureTotal}=T_{SecureCompute}\;T_{SecureUVS}+T_{w}$$
(9)$$T_{SecureUVS}=\sum_{i\in TNi}\frac{D_{UV\to i}}{S_{UV\to i}}$$
Here, $D_{UV\to i}$ denotes the distance between USNs and other entities, such as MMS, CH, and BS, deployed in the UVWSN. Thus, target legitimate USN and vulnerable sensor nodes are denoted as TNi, which include cluster head (CH) target legitimate USN, and vulnerable nodes used to collect secure data. Moreover, $S_{UV\to i}$ is the average moving speed the USN uses from the present position to the target legitimate USN and vulnerable sensor nodes, denoted as TNi position.

## 4. Simulation Setup, Result Analysis, and Discussion

For the simulation setup of the proposed SDAA protocol models, we simulate an underwater vehicular wireless sensor network (UVWSN) scenario. We simulated this scenario using low-power radios. These radios include a high asymmetrical coverage ratio communication range and stochastic link features. Simulating UVWSN application occurs in realistic results scenarios.

Therefore, UVWSN propagation utilizing USN (underwater sensor node) channel model is simulated through the proposed SDAA protocol. The simulation was developed and implemented in the network using the Omnet++ network simulation tool. We used IEEE 802.15.4 low power UVWSN compatible medium access control protocol and considered using an area size 1400 × 1400 square meters. The simulation time is 600 s, and 250 sensor nodes are used in the entire simulation area.

We chose 90% of sensor nodes to be static USNs (underwater sensor nodes), and the remaining 10% are considered mobile USNs that could move due to water waves and other physical interference. We generated a malicious node attacker (MNA) scenario for 1–18 attackers to assess and determine the effectiveness of the proposed SDAA protocol methods. The proposed SDAA protocol compares with other state-of-the-art similar types of approaches, including self-sustaining, efficient, and forward-secure cryptographic (SEFSC) secure data aggregation (SDA) and energy-efficient and secure transmission (EEST), which were deployed in the UVWSN in similar simulation condition.

The simulator parameters are shown in Table 3. Based upon this, we observed that similar scenarios of the simulation parameters are used throughout the investigation, which have been generated for the proposed SDAA protocol and the other competing standard secure MAC protocols, including SFSC, SDA, and the EEST. This is used to evaluate the effectiveness of the protocols. Based on the simulation results, we determine the following performance metrics.
End-to-end delayReliability ratioDrop packageEnergy consumption

**Table 3 sensors-23-05270-t003:** Simulation Parameters.

Parameters	Parameters Value
Number of nodes	250
Simulation time	600 s
Pause time	5 s
MAC protocols compared	SFSC, SDA, EEST, and SDAA
Traffic rate generated	60 Kb/s
Attackers generated	1 to 18
Propagation method used	Two ray ground
Antenna used	Omni antenna
The initial energy of each node	50 J
Received power	0.8 W
Transmission power	2.5 W
Mobile sink location	(300, 400)

### 4.1. End-to-End Delay

We analyze and discuss the end-to-end delay of the simulation result of the proposed SDAA protocols and compare it to the other standard secure MAC protocols, including SFSC, SDA, and EEST, for determining the network’s end-to-end (E2E) delay. Thus, the E2E delay of the network is determined by the total time data takes to reach base station (BS). This includes encryption time at underwater sensor nodes (USN), aggregation time at CH, transmission time from CH to BS, and decryption time at BS. Figure 4 depicts the results of the end-to-end delay for all the approaches. When malicious attackers are increased from 1 to 18, as shown in Figure 4, the delay of the proposed SDAA protocol increases from 0.017 to 0.039. However, the delay of the SDA approach increases from 0.018 to 0.043, the delay of EEST increases from 0.018 to 0.047, and the delay of SEFSC increases from 0.018 to 0.048. Hence, the delay of the SDAA protocol is 18.2% lesser than other contending approaches.

### 4.2. Average Data Reliability Ratio

In the simulation result, we determined the average data reliability ratio of the proposed SDAA protocol. We compared it with the other standard secure MAC protocols, including SFSC, SDA, and EEST. The average data reliability ratio in the network is the ratio of data packets received successfully to the total number of packets transmitted in the network. The average data reliability ratio reflects the efficiency and reliability of the network. Figure 5 depicts the results of the data reliability ratio for all the approaches. From Figure 5, as attackers increased from 1 to 18, the average delivery ratio of the SDAA protocol decreased from 100% to 98.4%, and the delivery ratio of EEST decreased from 100% to 74.3%. The delivery ratio of SEFSC decreases from 100% to 80%, and the delivery ratio of SDA drops from 100% to 88.2%. Hence, the proposed SDAA protocol gets a 10.2% to 24.1% higher delivery ratio than other contending approaches.

### 4.3. Average Packet Drop

We analyzed the simulation result of the proposed SDAA protocol and compared it with the other contending standard secure MAC protocols, including SFSC, SDA, and EEST. Based on the result of the average packet drop in the network due to malicious node attacker (MNA) in the network. The average packet drop in the network is determined as the number of packets dropped due to MNA attacks experienced in the network. Figure 6 depicts the results of the average packet drop for the proposed SDAA approach and the other contending approaches. As depicted in Figure 6, when the number of MNA attackers increases from 1 to 18, the packet drops in the network were observed at 34 with the proposed SDAA protocol and 36, 41, and 44 for SEFSC, EEST, and SDA, respectively. Hence, the packet drop of the proposed SDAA protocol is 4–20% lesser than other competing approaches. The reason for fewer average packet drops for our proposed SDAA protocol is the result of the robust authentication method and secure aggregated data optimized method deployed in CBND, as compared to the other contending approaches, such as SEFSC, SDA, and EEST, which did not utilize CBND and secure authentication methods in their networks.

### 4.4. Average Energy Consumption

We determined the average energy consumption of the proposed SDAA protocol. We compared it with the other contending standard secure MAC protocols, including SEFSC, SDA, and EEST, based on the simulation result obtained in the network. The average energy consumption in the presence of a malicious node attacker (MNA)of the network is determined as the amount of energy consumed by the underwater sensor nodes (USN) during data transmissions. This is expressed as the average energy consumption of all the network USNs and the mobile nodes during the simulation. Figure 7 depicts the average energy consumption for all the approaches.

The trend in the graph shows that more energy is consumed when the number of attackers increases from 1 to 18. Based upon this, the proposed SDAA protocol consumes 30 J, and the energy consumption of SEFSC, SDA, and EEST is 37, 42, and 44, respectively. Hence, the SDAA protocol consumes 7–14 J less energy than other competing approaches.

## 5. Conclusions

In this research, we have proposed secured data aggregation and authentication-optimized methods in a cluster-based based network design (CBND) deployed in an underwater vehicular wireless sensor network termed UVWSN. UVWSN utilizes application monitoring and the inspecting of a vehicle or ship structures for the navigational system in the ocean. Performance metrics of the network depend on reliability, energy efficiency, and the average packet drop of the USNs (underwater sensor nodes) based on multiple mobile sinks (MMSs) deployed in the CBND UVWSN. Performance metrics assessment depends on malicious node attacker (MNA) detection in the network. Performance metric measurements, including energy efficiency and average packet drop evaluation in our proposed SDAA protocols, were determined and compared with the other contending standard secure MAC protocols, such as SEFSC, SDA, and EEST, deployed in similar simulation conditions in the network.

Models of the proposed SDAA protocols were developed. They were deployed in the UVWSN scenario, which evaluates the network in the same condition as the other contending secure MAC protocol for measuring reliability, packet drop, delay, and energy efficiency in the network. In analyzing the simulation results based on the model developed for the proposed SDAA protocol, the proposed SDAA protocol guarantees higher data security in terms of reliability, packet drop, end-to-end delay, and average energy consumption. Therefore, the proposed SDAA protocol achieved higher energy efficiency of 30 J, less packet drop ratio, and less delay, as compared to the other contending standard secure MAC approaches, including EEST, which had 44 J, SEFSC had 37 J, and SDA had 42 J.

For future work, we plan to improve the proposed SDAA protocol by analyzing and detecting specific types of attacks, such as DDoS and collision attacks, at the MAC layer.

## Figures and Tables

**Figure 1 sensors-23-05270-f001:**
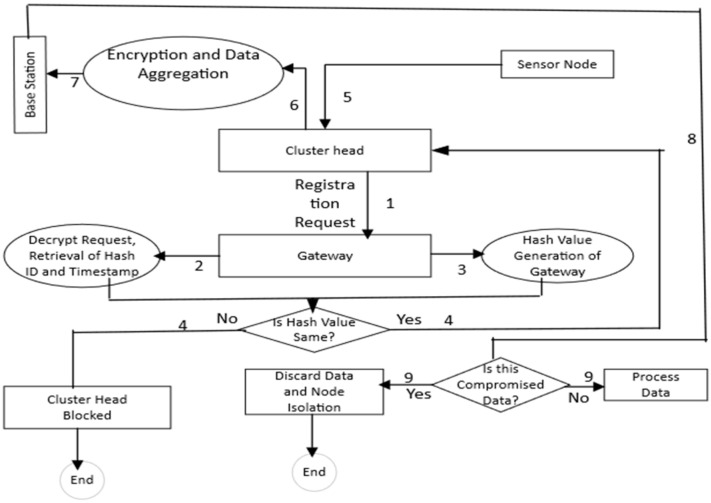
Aggregation and authentication processes in CBND.

**Figure 2 sensors-23-05270-f002:**
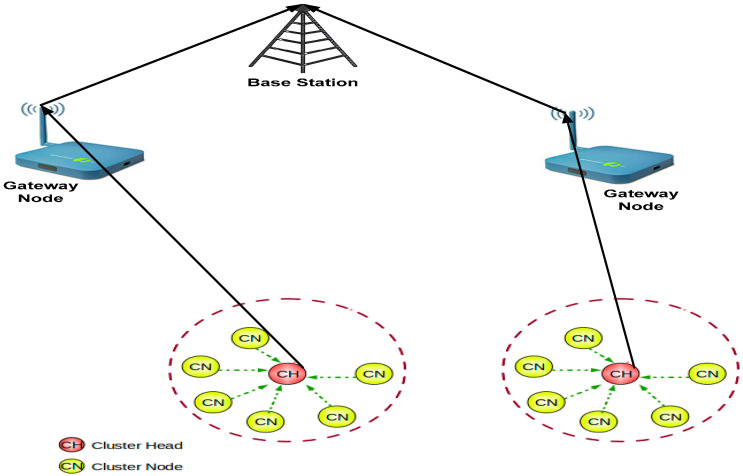
System model for secure data aggregation and authentication in cluster-based network design.

**Figure 3 sensors-23-05270-f003:**
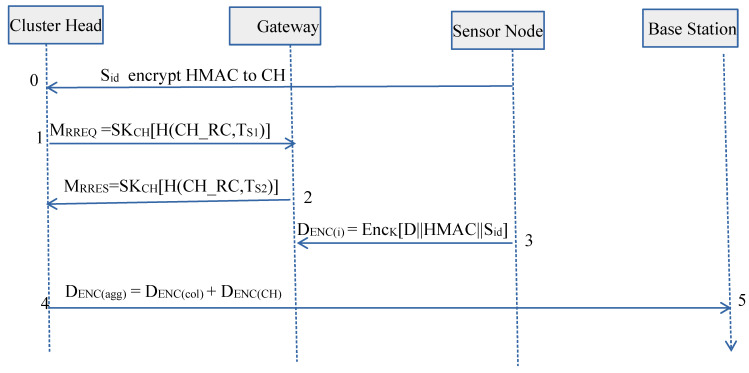
Process Flow of SDAA.

**Figure 4 sensors-23-05270-f004:**
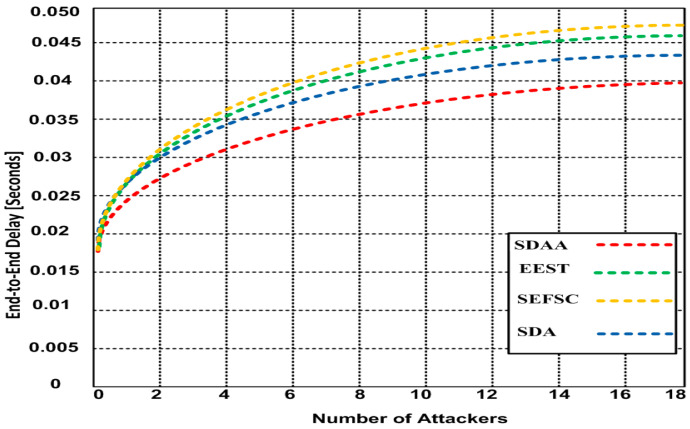
Average end-to-end delay with a varying number of attackers.

**Figure 5 sensors-23-05270-f005:**
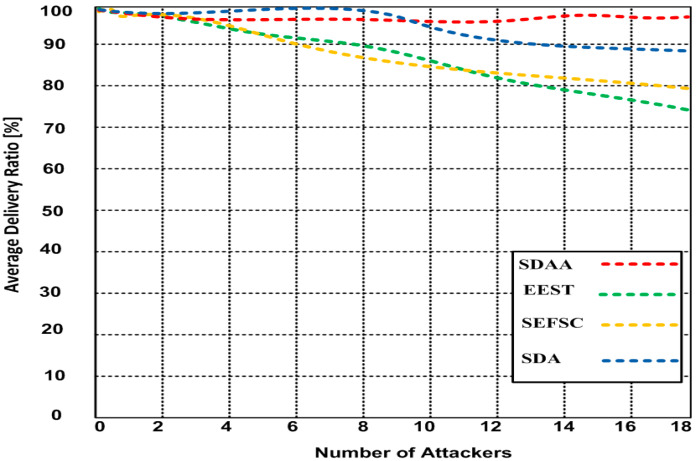
Average delivery ratio varying number of attackers.

**Figure 6 sensors-23-05270-f006:**
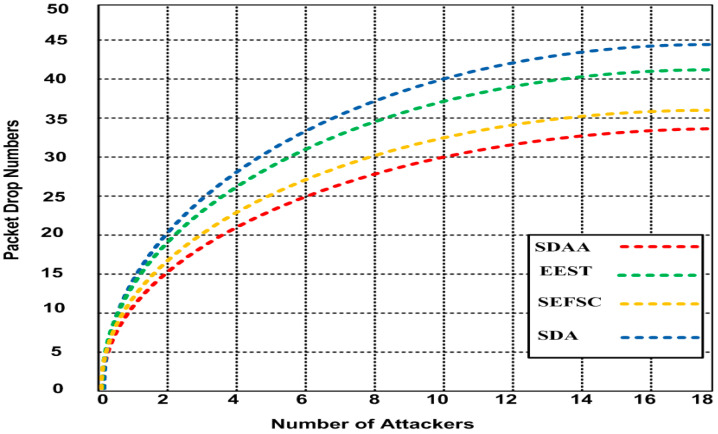
Average dropped packets for SDAA and comparison with various attackers.

**Figure 7 sensors-23-05270-f007:**
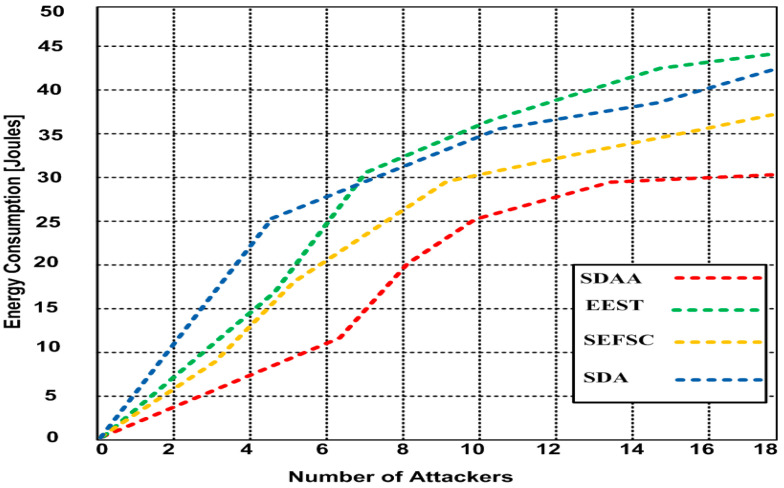
Average energy consumption of SDAA and competing approaches with various attackers.

**Table 1 sensors-23-05270-t001:** Notation and Description of Algorithm 1.

Notations	Description
*SK_CH_*	A secret key generated by CH
*PK_CH_*	Public key of CH
*CH_id_*	The ID of the CH
*GW_id_*	The ID of the gateway
*CH_RR_*	CH registration request
*CH_RC_*	CH registration confirmation
*M_RREQ_*	Registration request message
*M_RRES_*	Registration response message
*H*	Hash function
*T* _*s*1_	Timestamp at which the request was generated at CH
*T* _*s*2_	Timestamp at which the reply was generated at GW
*SK_GW_*	Secret key of gateway
*PK_GW_*	Public key of gateway

## Data Availability

Available upon request.

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
