# Peer review of "SDAA: Secure Data Aggregation and Authentication Using Multiple Sinks in Cluster-Based Underwater Vehicular Wireless Sensor Network"

_sensors, 2023, doi:10.3390/s23115270_

Round 1
Reviewer 1 Report (Previous Reviewer 3)
1. The proposed method in this research investigates UWSN combined with UV optimized together as an underwater vehicular wireless network (UVWSN) that can completely detect malicious node attacks (MNA) from the network. In this proposed SDAA protocol, the cluster head (CH) is authenticated through the gateway (GW) and the base station (BS) to guarantee that legitimate USN oversees all clusters deployed in the UVWSN are securely authenticated for providing trustworthiness/privacy
2. Pages 7, 21, and 22 should be revised.
3. In the figure 1, aggregation and authentication processes in CBND should be demonstrated in detail.
4. In the figure 3, process Flow of SDAA should be demonstrated in detail.
5. The authors are suggested to highlight the contributions of the proposed work, compared to the prior works. A detailed discussion about prior works are suggested to add.
6. Some future works are suggested to discuss to give an enlighten to the readers.
7. The manuscript has 26 pages; the number of the pages should be decreased.
8. Revise the English thoroughly before re-submission.
Extensive editing of English language required.
Author Response
The authors wish to sincerely thank the reviewers for their constructive criticism on the Manuscript and have tried to answer the reviewers comments to the best ability.
Thank you so much for your good work.
2 Pages 7, 21, and 22 should be revised.
Page 7-A the page has been thoroughly revised with a lot of information minimized. Thank you
Page 21 and 22 –
In page 21 spacing has been created to separate information for content clarity and page reduction.
In page 22 a lot of information has been revised to suit content well
- In the figure 1, aggregation and authentication processes in CBND should be demonstrated in detail.
In figure one end of process components for SDAA have been repositioned to clarify it in detail
- In the figure 3, process Flow of SDAA should be demonstrated in detail.
In Figure 3 has been redrawn to best ability and new component of flow diagram has been added. Thank you very much .
- The authors are suggested to highlight the contributions of the proposed work, compared to the prior works. A detailed discussion about prior works are suggested to add.
Authors have added some information to give detail but short (due to page reduction) discussion of prior works.
- Some future works are suggested to discuss to give an enlighten to the readers.
Authors have added clear future work for readers enlightenment . Thank you
- The manuscript has 26 pages; the number of the pages should be decreased.
The manuscript pages have been reduced from 26 to 24 (excluding reference section). Thank you
- Revise the English thoroughly before re-submission.
The whole manuscript has been thoroughly revised in the contents. Thank you.
Reviewer 2 Report (Previous Reviewer 1)
I have no further comment.
Minor editing of English language is required.
Author Response
The whole manuscript has been thoroughly revised in the contents for English. Thank you.
Round 2
Reviewer 1 Report (Previous Reviewer 3)
no further comment.
Minor editing of English language required.
Author Response
Manuscript has been completely edited for minor english
This manuscript is a resubmission of an earlier submission. The following is a list of the peer review reports and author responses from that submission.
Round 1
Reviewer 1 Report
This paper considers secure data aggregation and authentication design for underwater sensor networks. My comments are given as follows.
1. The readability of this work is poor. It is easy to detect lots of grammatical errors. Extensive revision is strongly required.
2. The novelty of this work is not clear. Need to highlight.
(a) Employment of multiple sink nodes can not be defined as a contribution.
(b) The used encryption schemes are well researched.
3. Authentication is only a branch of security. To enrich the background, some popular security related designs should be mentioned in the introduction part, such as privacy design: Fundamentals of physical layer anonymous communications: sender detection and anonymous precoding, IEEE Trans. Wireless Commun.
4. Please carefully proofread the manuscript before submission, such as inconsistent font size, unclear figures, different line space, undefined variable or abbreviations. The current organization of this work is very poor.
Reviewer 2 Report
This paper deals with very interesting and modern topic, which is in the focus of research community. However, paper has significant flaws, which make it difficult to read and follow, making it inappropriate for publishing in present form in the highly ranked journal, as Sensors.
Since some of the presented results are promising and seems very good, I suggest to authors to rewrite most of the text in order to be more informative and easier to follow. Maybe, one of the reasons for this opinion could be the style of writing in English, which makes this text hard to read.
I will give just some of my suggestions to authors in order to possibly improve this manuscript.
1. Introduction is confusingly written, and many things are repeated few times, such is VMNI phenomena. It is too long, and many parts could be transferred to the section “Related work”. Generally, Introduction does not reflect purpose of the introductory section.
2. Many sentences are just uninformative. For example: “Understanding what a contention-free MAC protocol is in WSN brings the network from an arbitrary state to a collision-free, stable state” (section 2.2). How “understanding” of anything can bring network to any state?!
3. What is the meaning of the following statement: “MAC protocol is used to transmit messages without collision to ascertain what happens in the stable state at the stable state contention-free”? This is another example, among many of them, of unclear statements/sentences.
4. In the beginning of section 3, authors mention some “lines” describing utilizing two steps in secure data aggregation and authentication (lines 1-4 and 5-9). I guess they meant labels in the figure 1, which is given AFTER this referring to the “lines”. Readers must not guess, it has to be clear and straight forward.
5. Aggregation and authentication processes depicted in figure 1 should be better explained.
6. There are two “if boxes” In figure 1 which have “yes” option, “no” option, and third option leading to “end”. Which is this third option, if not yes or no?!
7. Figures are generally larger than necessary, especially figure 2.
8. Table 1, with the notation and description, should be placed before Algorithm 1. By the way, there are two “table 1” in the manuscript!
9. Please, check numeration in Algorithm 2. There are some empty lines, and some numbers are missing.
10. General remark: algorithms are poorly explained. Specially, 4, 5, and 6.
11. Process flow in figure 3 seems incomplete.
12. Table 2 should also be given before the corresponding algorithm.
13. In section 3.4, Pr is probability of what?
14. There is typo in equation (2). Generally, equations are not clearly presented, many of them with suspicious items (like dots in (3), for example), or looks strange (such eq. (7), for example).
15. Last sentence in page 18 looks unfinished and uncomplete.
16. In section 3.10 authors say: “We investigate and determine the delay and energy efficiency in [36] as below”. But, reference 36 is not their work?!
17. At the beginning of the section 4 authors say: “Simulation results could be marginally different from realistic results, but they can give an almost similar result”. What is the basement for this claim? Some reference, or you have experimental and simulation results and compared them? Or something else?
18. Authors considered area size 1400 x 1400 square meters, simulation time is 600 seconds, and 250 sensor nodes are randomly deployed in the entire simulation area. What is the reason to choose exactly these parameters?
19. Authors should give some information about simulator used in this research.
20. In figure 5, diagram of SDAA looks almost constant with increasing number of attackers. There is not any comment of this behavior. Just opposite: authors claim that parameter decreases which is not quite true.
21. As presented and defined in the text, parameters “Average Data Reliability Ratio” and “Average Packet Drop” are related to each other. However, results does not support this. It means that these are not related (poorly defined and explained), or results are not accurate.
22. Conclusion is also not addressed appropriately. There is no need to explain models and algorithms in the conclusion. Only results for energy efficiency are commented, and even them only for 18 attackers.
Reviewer 3 Report
1. The authors introduce a secure data aggregation and authentication (SDAA) method for cluster-based UWSN and UVS systems. In this method, the cluster head is authenticated through the gateway to guarantee that legitimate sensor nodes oversee all the clusters.
2. Pages 1, 3, 4, 5, 17, and 18 should be revised.
3. In the figure 1, Aggregation and authentication processes should be demonstrated in detail.
4. The authors are suggested to highlight the contributions of the proposed work, compared to the prior works. A detailed discussion about prior works are suggested to add.
5. Some future works are suggested to discuss to give an enlighten to the readers.
6. The manuscript has 28 pages; the number of the pages should be decreased.
7. The manuscript can be resubmitted.
8. Revise the English thoroughly before re-submission.